# Schools of all backgrounds can do physics research: On the accessibility and equity of the PRiSE approach to independent research projects

Martin O. Archer[1,*]

[1]School of Physics and Astronomy, Queen Mary University of London, London, UK
[*]now at: Space and Atmospheric Physics, Department of Physics, Imperial College London, London, UK

**Correspondence:** Martin O. Archer
(martin@martinarcher.co.uk)

**Abstract.** Societal biases are a major issue in school students' access to and interaction with science. Schools engagement programmes in science from universities, like independent research projects, which could try and tackle these problems are, however, often inequitable. We evaluate these concerns applied to one such programme, 'Physics Research in School Environments' (PRiSE), which features projects in space science, astronomy, and particle physics. Comparing the schools involved with PRiSE to those of other similar schemes and UK national statistics, we find that PRiSE has engaged a much more diverse set of schools with significantly more disadvantaged groups than is typical. While drop-off occurs within the protracted programme, we find no evidence of systematic biases present. The majority of schools that complete projects return for multiple years of the programme, with this repeated buy-in from schools again being unpatterned by typical societal inequalities. Therefore, schools' ability to succeed at independent research projects appears independent of background within the PRiSE framework. Qualitative feedback from teachers show that the diversity and equity of the programme, which they attribute to the level of support offered through PRiSE's framework, is valued and they have highlighted further ways of making the projects potentially even more accessible. Researcher-involvement, uncommon in many other programmes, along with teacher engagement and communication are found to be key elements to success in independent research projects overall.

## 1 Introduction

It has long been the case that the Science, Technology, Engineering, and Mathematics (STEM) sectors have shown, both within higher education and in the workforce, systemic biases against women, other non-male genders, those from ethnic-minorities, and socially-disadvantaged groups (e.g. Campaign for Science and Engineering, 2014). Inequalities are present even at the secondary/high school level, where students from under-represented or disadvantaged backgrounds, despite being interested in science, have fewer opportunities to engage with science both inside and outside of school (Hamlyn et al., 2020). These societal issues constitute major inequities on young people that influence their opportunities, self-perception, and ultimately subject/career choices. It is therefore important that STEM engagement programmes aim to equitably include students from these groups, taking into account all of the factors that might support or prevent their engagement.

Independent research projects are opportunities enabling school students to conduct open-ended investigations in science. While some independent research project initiatives specifically target traditionally under-represented groups (with many of these being in the USA), a recent global review (Bennett et al., 2016, 2018) found that in general there are equity issues relating to participation in such projects — they are relevant to all young people, but only a small minority are able to access them. The authors note that despite emerging evidence that independent research projects can result in improved engagement and attitudes towards science amongst students from under-represented groups, further work is required to more fully explore the potential benefits of independent research projects on them.

'Physics Research in School Environments' (PRiSE) is a framework for independent research projects for 14–18 year-old school students that are based around cutting-edge physics research and mentored by active researchers (Archer, 2017; Archer et al., 2020). Unlike some citizen science initiatives with schools, which due to their focus on answering specific science questions can sometimes result in an inauthentic research experience focused around crowdsourcing (Bonney et al., 2009, 2016; Shah and Martinez, 2016), PRiSE as a 'research in schools' programme was devised in an audience-focused way, with the benefits to participants being of primary importance (discussed in more detail in Archer et al., 2020). Thus far the four PRiSE projects summarised in Table 1 have been developed at Queen Mary University of London (QMUL) since 2014 and the framework is now being adopted by other institutions who are developing their own projects applied to their specific areas of physics research. The programme aims to equitably include significant numbers of students from demographic groups which are under-represented in higher education and STEM. Projects run from the start of the UK academic year in September to just before the spring/Easter break in March, a duration of approximately six months. The role of the teacher in these projects is chiefly one of encouraging their students to persist, providing what advice they can, and then communicating with the university. Teachers are not expected to fully manage the projects, which is why numerous modes of support are provided from active researchers who have the expertise and skills in the areas of each project. This support offered to students and teachers comes in the form of a suite of bespoke resources along with the following intervention stages each year:

- **Assignment (Jun–Jul)** Teachers sign their school up for a PRiSE project and are informed of the outcome before the summer break.

- **Kick-off (Sep–Oct)** An introductory talk and hands-on workshop, either in-school or as an evening event on university campus.

- **Visit (Dec–Feb)** Researchers visit the schools to mentor students on their project work.

- **Comments (Mar)** Researchers provide comments on students' draft presentations near the end of the project.

- **Conference (Mar)** Students present their project work as either posters or talks at a student conference held on campus and attended by teachers, family, and researchers.

and any further ad hoc communications as needed on an individual school basis. Evaluation has shown that all of these elements of support are almost equally important and necessary in the eyes of students and teachers (Archer et al., 2020). This

| Project | Abbreviation | Years | Field | Description |
|---------|-------------|-------|-------|-------------|
| Scintillator Cosmic Ray Experiments into Atmospheric Muons | SCREAM | 2014–2020 | Cosmic Rays | Scintillator – Photomuliplier Tube detector usage |
| Magnetospheric Undulations Sonified Incorporating Citizen Scientists | MUSICS | 2015–2020 | Magnetospheric Physics | Listening to ultra-low frequency waves and analysing in audio software |
| Planet Hunting with Python | PHwP | 2016–2020 | Exoplanetary Transits | Learning computer programming, applying this to NASA Kepler and TESS data |
| ATLAS Open Data | ATLAS | 2017–2020 | Particle Physics | Interacting through online tool with LHC statistical data on particle collisions |

**Table 1.** A summary of the existing PRiSE projects at QMUL.

paper assesses whether the approach taken and level of support provided by PRiSE enables schools from all backgrounds to participate and succeed in independent research projects. Section 2 investigates the diversity of schools that have participated in PRiSE, benchmarking them against UK national statistics as well as schools involved in other similar programmes. We then investigate retention of the schools in PRiSE, both within each academic year and across multiple years, in section 3. Finally, feedback from teachers relating to diversity, accessibility and equity are presented in section 4. Impacts of the programme upon students, teachers and schools and whether these are potentially affected by background are discussed in a companion paper (Archer and DeWitt, 2020).

## 2    Participation

As of March 2020, 67 schools have been involved in PRiSE. A full list of (anonymised) schools is given in Appendix A. Figure 1 demonstrates that these schools (blue) have been fairly broadly spread across Greater London rather than being focused solely around Queen Mary (red). Schools targeting has been limited to London to enable researchers to build relationships with the schools via their in-person interactions throughout the 6 month projects. Most schools have participated directly with Queen Mary, though we note that some have been involved as a partnership of local schools (there have been 8 partnerships across 22 schools, listed in Appendix A). Such partnerships could provide an additional support network to students and teachers as well as making interventions more efficient for researchers. However, we have found these partnerships to have been somewhat hit-or-miss so far within PRiSE — while kick-off events with all partner schools present have typically worked, following this the schools have not always worked with their partners on the projects. Further investigation is required to understand what makes these partnerships work.

Here we evaluate the diversity of schools engaged in PRiSE. We limit this analysis to publicly available data concerning the schools and their local areas, and we did not collect any protected characteristics (such as gender or race) or sensitive information (such as socio-economic background) from the students involved. This was done for both ethical and practical reasons, bearing in mind that this is a schools engagement programme delivered and evaluated by physics researchers and not

an educational research project in and of itself. For example, it was deemed that requiring students or their teachers to provide protected or sensitive information upfront would have risked some students, or indeed entire schools, declining to participate. This limits the conclusions that can be made to only the school-level. However, it has been recognised that the clustering of

80 students within schools results in students within the same school having more in common with each other than with students in different schools, an important consideration in the uptake of post-compulsory physics education for example (Gill and Bell, 2011). While multilevel models could account for this hierarchy, this is beyond the scope of what is practical for PRiSE. We note that schools typically involve entire (or significant fractions of) cohorts of A-level physics students in PRiSE and so while we have no indication that PRiSE students differ in any substantive way from their schools' wider student-base, we cannot

rule out that they may not necessarily be representative. Finally, since one of the aims of PRiSE is to impact on teachers' practice and schools' STEM environments, school-level considerations are valuable in this context regardless of the specific characteristics of the students engaged in PRiSE.

We benchmark school-level data against UK national statistics as well as schools listed on the websites of two other similar UK-based programmes of research-based physics projects for schools, IRIS (2018, $n = 178$) and HiSPARC (2018, $n = 22$).

While we also looked at schools involved with ORBYTS (2019) who specifically mention targeting disadvantaged groups, finding very similar results to PRiSE, with only 17 schools listed there is limited scope for detailed statistical comparison and so we have omitted this programme here. We make no comment on the reasons behind the makeup of schools involved with different programmes, since this would require specific qualitative research into how each programme's provision model and targeting affects participation. Information about schools was first obtained from the 'Get information about schools' database,

formerly known (and henceforth referred to in this paper) as Edubase (Department for Education, 2018). For more information on the UK schools system, please see Appendix B. While all PRiSE and HiSPARC schools could be found in Edubase, only 154 of the listed IRIS schools could be identified (based on UK postcodes).

Figure 2 shows the makeup of school categories (explained in Appendix B) from this database across the three programmes, showing little overall difference between them — in a chi-squared test of independence $\chi^2(8) = 8.45$ corresponding to $p =$

0.391. While none of the differences between the programmes are strictly statistically significant (using a difference in binomial proportions test) due to the relatively small numbers of schools compared to the population, it should be noted that IRIS features proportionally more independent schools than PRiSE ($+0.10$ absolute and $1.46\times$ relative, $p = 0.107$), HiSPARC involves more academies than PRiSE ($+0.16$ and $1.40\times$, $p = 0.260$), and PRiSE works with more local authority maintained schools than both HiSPARC ($+0.08$ and $1.52\times$, $p = 0.418$) and IRIS ($+0.09$ and $1.60\times$, $p = 0.147$) . However, it is clear that none of

the programmes are truly representative of all schools nationally by category. This is also the case when looking at schools' admissions policies (again see Appendix B for further background). The $10 \pm 4\%$ of selective schools in PRiSE is more than the $1\%$ nationally listed in Edubase, though we note that both HiSPARC and IRIS feature even higher proportions of selective schools than PRiSE at $14 \pm 9\%$ ($p = 0.342$) and $24 \pm 4\%$ ($p = 0.007$) respectively, where the uncertainties refer to the standard (i.e. 68%) Clopper and Pearson (1934) confidence interval in a binomial proportion. To address the imbalances in school

categories and admissions policies of PRiSE schools, as of 2019 we have implemented a policy that all independent and selective schools must partner with local state or girls' schools, including them in their project work. While this was something

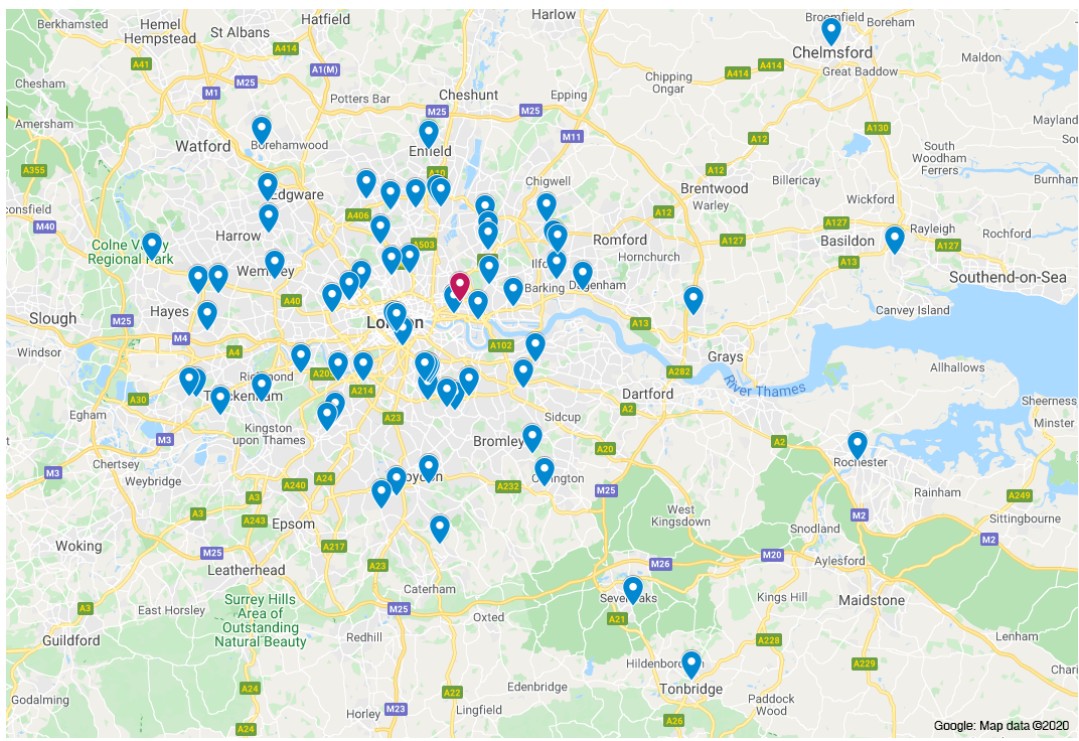

**Figure 1.** Map of all schools involved with PRiSE (blue) with Queen Mary University of London's location shown in red.

which some independent schools had voluntarily been doing previously, we have encountered some resistance to this policy by a number of independent schools. Schools which refused the policy were not allowed to participate, even if they had worked with us previously. While several other schools agreed to the principle and tried to implement partnerships, some failed to do so due to them not being able to draw from existing local school partnerships, limited time from the application to the summer holidays, and poor communication between teachers at different schools. These schools were allowed to participate, with the expectation that they put further effort in to establish these partnerships ready for the next academic year, which they seemed willing to do.

Beyond school type and admissions policy we look at several other metrics for the backgrounds of the schools' students. We detail in Appendix C how we have combined various datasets in order to assess these. For PRiSE schools these methods results in two different metric values for each school — one covering the school's full catchment area (the entire area from which they draw students) and another purely pertaining to the school's local census area (the immediate area surrounding the school's location). We consider the full catchment area data to be more reflective of a school's student base compared to the local data. For schools outside of London, however, we only have access to the local data. Note that the two types of data can result in rather different values for a school despite the underlying distributions across all London schools being similar (see Appendix C for further discussion). In Figure 3 distributions of the gathered metrics are shown in two formats. Top panels display boxplots depicting quantiles of the metric. Bottom panels depict kernel density estimates of the continuous probability

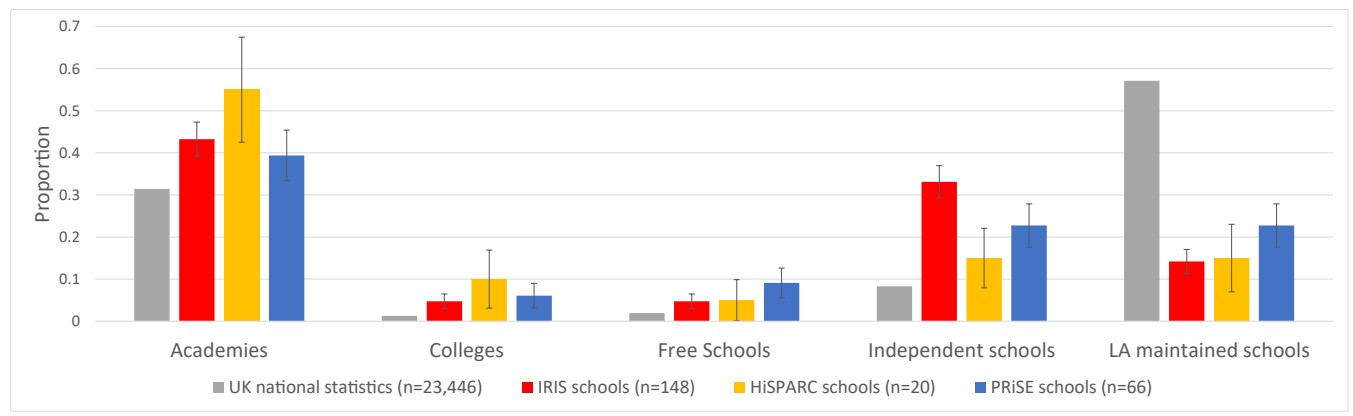

**Figure 2.** Distribution of school categories involved in IRIS (red), HiSPARC (yellow) and PRiSE (blue) projects compared to UK national statistics (grey). Error bars denote standard Clopper and Pearson (1934) intervals.

distributions, where Gaussian kernels of optimal bandwidth from the Silverman (1986) rule have been applied to each dataset. Standard confidence intervals are estimated by bootstrapping 1,000 random realisations of the data (Efron and Tibshirani, 130   1993), taking quantiles of their resulting kernel density estimates (with the bandwidth fixed from before).

The first metric we consider is the percentage of students eligible for free school meals, an often-used widening participation criterion in the UK which can be found in Edubase. Free school meals are a statutory benefit available to school-aged children from families who receive other income-assessed government benefits and can be used as a proxy of the economic status of a school's students. Figure 3a shows that both HiSPARC and IRIS schools feature considerably lower free schools meals 135   percentages than the national statistics in terms of the location (e.g. median), scale (e.g. interquartile range), and shape (e.g. tail heaviness) of their distributions. In contrast, the kernel density estimate for PRiSE schools appear to tend towards higher percentages and be somewhat broader. We perform Wilcoxon rank-sum tests, which test whether one sample is stochastically greater than the other (often interpreted as a difference in medians) since it is more conservative and suffers from fewer assumptions (e.g. normality, interval-scaling) than two-sample t-tests (Hollander and Wolfe, 1999; Gibbons and Chakraborti, 140   2011). These tests, however, reveal that in terms of quantiles PRiSE schools are merely consistent with the national distribution (an absolute $+3\%$ and relative $1.26\times$ difference in medians, $p = 0.708$), as is also evident from the boxplots. This means that PRiSE is serving schools with considerably more disadvantaged students than both HiSPARC ($+7\%$ and $2.09\times$, $p = 0.033$) and IRIS ($+8\%$ and $2.18\times$, $p = 2 \times 10^{-4}$) in this regard.

The second metric considered is the higher education participation rate (Figure 3b), which measures how likely young people 145   are to go on to higher education (e.g. university) based on where they live (Office for Students, 2018). All the programmes considered involve schools with students from areas with greater participation in higher education than is representative of the entire country, but they are largely similar to one another. This is not surprising for PRiSE given it is limited to London, since it has been noted that young people across London are generally more likely to access higher education than those elsewhere in the UK (Office for Students, 2018). However, the fact that PRiSE's results are similar to the two national programmes is

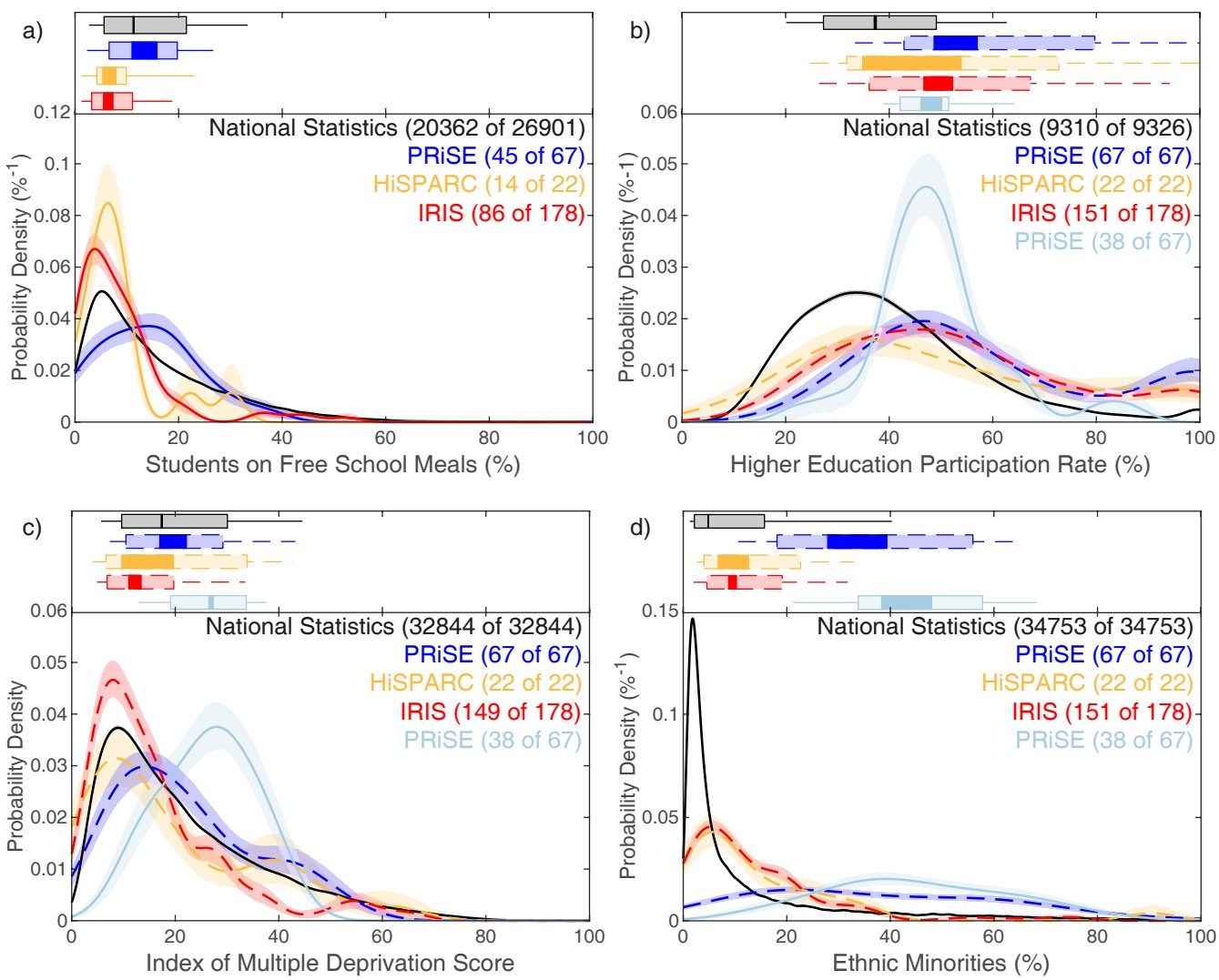

**Figure 3.** Distributions of a) students on free school meals, b) higher education participation rate, c) index of multiple deprivation score, and d) ethnic minorities. Boxplots in the top panels have whiskers covering 10–90%, boxes spanning 25–75%, and bands depicting the standard confidence interval in the median. Lower panels show kernel density estimates along with bootstrapped standard confidence intervals. Dashed lines indicate only local data is used, which does not cover the schools' full catchment areas.

perhaps surprising as HiSPARC lists no London schools and only $17 \pm 3\%$ of IRIS schools are in the Greater London area. Therefore one might expect these two programmes to have markedly lower participation rates in higher education than PRiSE purely due to this fact, which is not the case. Given in-person interactions between researchers and schools is a critical part of the PRiSE model, the geographical reach of the Queen Mary programme will always be limited to London and thus it is difficult to effect much change on the higher education participation rate. The expansion of PRiSE to universities in other areas however may help address this in future.

The third metric used is the index of multiple deprivation (Ministry of Housing, Communities & Local Government, 2015), a UK government qualitative study of deprived areas in English local councils by: income; employment; health deprivation and disability; education, skills, and training; barriers to housing and services; crime; and living environment. Here we use the index of multiple deprivation scores (Figure 3c), where higher scores indicate more deprivation. Averaged over each PRiSE school's catchment area, these scores are considerably higher than the national statistics (the median is $+9.40$ and $1.54\times$ larger, $p = 7 \times 10^{-4}$). This difference, however, disappears when using the local proxy yielding simply a representative distribution ($+1.50$ and $1.09\times$, $p = 0.371$). In contrast, IRIS schools again clearly favour fewer disadvantaged students than PRiSE ($-6.56$ and $0.65\times$, $p = 4 \times 10^{-4}$) and thus also the national statistics ($-5.07$ and $0.71\times$, $p = 2 \times 10^{-6}$), whereas these differences are perhaps only marginal for HiSPARC due to small numbers ($-6.84$ and $0.64\times$, $p = 0.142$ compared to PRiSE; $-5.35$ and $0.69\times$, $p = 0.288$ compared to national statistics).

The final metric used is the percentage of ethnic minorities (Figure 3d) taken from census area data. The PRiSE programme features a very broad distribution with much greater percentages of ethnic minorities than both the 13% of all people from ethnic minorities across the UK (Office of National Statistics, 2011) and those from the areas covered by the other programmes. This is simply due to the fact that London is the most ethnically diverse region in the country though. The distributions for IRIS and HiSPARC are very similar to one another and while their distributions' location parameters are just less than the overall national statistic, they both feature greater ethnic diversity than compared to the full distribution across all census areas (in the case of HiSPARC this is not strictly statistically significant at $+4\%$ absolute and $1.79\times$ relative, $p = 0.056$, likely due to small number statistics).

In terms of gender balance, a similar school/area-level approach would not capture the known bias present in physics where only 22% of A-Level physics students are female (Institute of Physics, 2018). Therefore, at kick-off meetings the percentage of young women or girls involved in PRiSE at each school were observed (accurate to the nearest 10%). The median of these is 40% though there has been considerable variation in the gender balance amongst schools. We have worked with 11 girls' schools (compared to just 8 boys' schools) to help address gender balance across the programme. How this variation on a school-by-school basis compares to each schools' A-Level cohorts or across all schools nationally is unknown, as this data is not publicly available. IRIS, HiSPARC and ORBYTS have not yet reported on the gender variations in their programmes.

To summarise, PRiSE has to date engaged with a much more diverse set of schools with significantly more under-represented groups than other similar schemes and reflects national statistics in most measures, sometimes even featuring higher proportions of underserved groups. Some work, however, needs to be done for PRiSE to be more representative in terms of schools' categories and admissions policies, which is currently being addressed. Future work could expand the evaluation of partici-

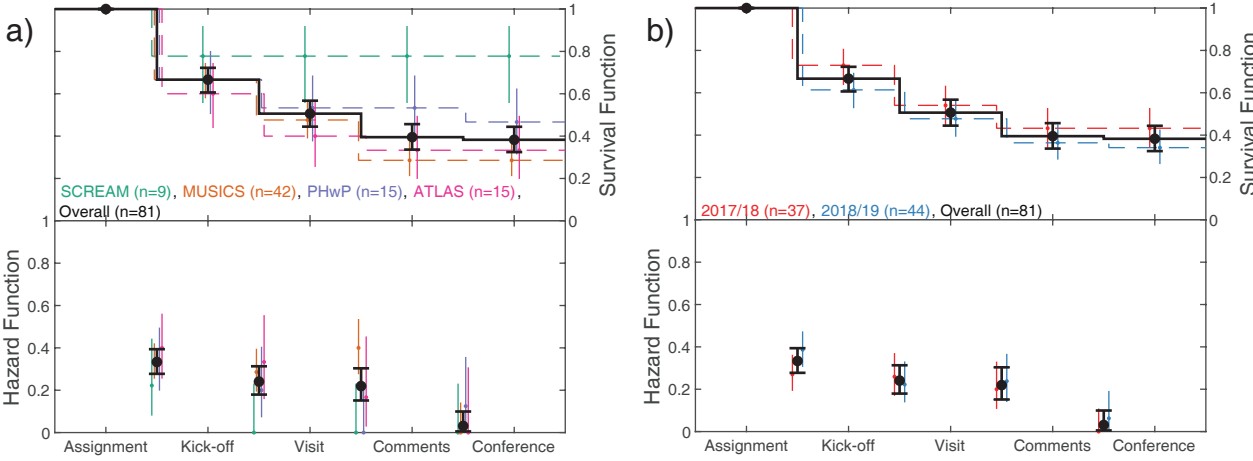

**Figure 4.** Schools' survival (top) and hazard (bottom) functions across interventions. Overall results in black, colours subdivide by a) project and b) year. Error bars denote standard Clopper and Pearson (1934) intervals.

pation to go beyond school/area-level metrics and investigate individual students' characteristics, however this would require funding to commission such research by social scientists along with the necessary ethical approval.

## 3 Retention

It is natural to expect some drop-off in participation of schools for a long-term and primarily extra-curricular programme. We employ the branch of statistics known as survival analysis (Miller, 1997) to quantify the retention of schools throughout the PRiSE programme. In particular we calculate both survival (the probability that a school is still involved with the programme at a certain stage) and hazard (the probability a school still involved at a certain stage will drop out at the next stage) functions.

### 3.1 Across interventions

First we assess the retention of schools across the intervention stages of the programme. This is done for academic years 2017/18 and 2018/19 only, since before this our data collecting was insufficient to track schools' retention throughout the different stages of the programme and in 2019/2020 the programme was disrupted by the COVID-19 pandemic just before the comments stage with the conference having to be postponed. In many cases in our results an assumption has been made about when schools may have dropped out, because in $68 \pm 12\%$ of schools which drop out after the kick-off stage the teachers do not inform us and simply no longer reply to our continuing emails (with little difference in this percentage across the different project stages). In these cases we assume that schools dropped out at the earliest point the teacher became unresponsive.

Figure 4 shows schools retention both by project (panel a) and year (panel b) across the different intervention stages. Overall there is a fairly uniform drop-off rate between the kick-off – visit – comments stages at $23 \pm 7\%$ with a slightly higher drop-off from assignment to kick-off ($33 \pm 5\%$). Schools still involved by the comments stage almost certainly attend the conference.

None of the other 'research in schools' programmes have yet reported on retention within their programmes and in general little research into retention within (particularly free) programmes of multiple STEM interventions with schools exists. However, we note that the figures from PRiSE are at least consistent another programme — those of the South East Physics Network (SEPnet) Connect Physics pilot (Hope-Stone Research, 2018). No overall differences across survival distributions in a logrank test (Machin et al., 2006) appear present by year ($\chi^2(1) = 0.60$, $p = 0.441$) or project ($\chi^2(3) = 4.28$, $p = 0.232$). When comparing different projects in the same year and the same projects in different years there are differences in the survival/hazard functions to those shown in Figure 4 though (see data in Appendix D). One interpretation of the results shown might be that projects where schools are lent equipment (i.e. SCREAM) are more likely to succeed. Past experience across SEPnet with the CERN@School IRIS project doesn't support this hypothesis, however, since almost all SEPnet target schools that were loaned detectors ended up not using them (D. Galliano, personal communication, 2018). To prevent equipment going unused, the SCREAM project has been made open only to schools that have successfully undertaken a different project with us previously, which likely plays a factor in the results presented. In general, projects based around expensive equipment are not scalable given funding limitations so would necessarily always have a limited reach.

While all intervention stages are offered to all schools, it should be noted that not all schools actively engaged in project work take advantage of them: $7 \pm 4\%$ don't attend a kick-off, $39 \pm 9\%$ do not schedule a researcher visit, and $47 \pm 10\%$ don't solicit comments on their work. Schools which don't solicit a researcher visit are much more likely to subsequently drop out ($41 \pm 14\%$) compared to those which do ($12 \pm 8\%$). This highlights the importance of researcher-involvement in the success of programmes, despite independent research projects in schools often not being supported by external mentors in general (Bennett et al., 2016, 2018). The other two intervention stages do not appear to be critical for schools' retention. Firstly, in the case of the kick-off, this suggests the resources provided are sufficient to still undertake project work. Secondly, while comments on draft presentations are valued by students and teachers, students are still able to produce work that can be presented at a conference without them.

In addition to equality in access, it is also important that programmes be equitable so that everyone has a fair chance of succeeding. We find no biases in schools' ability to successfully complete a year by school category ($\chi^2(4) = 3.21$, $p = 0.524$), free schools meals percentage ($+4\%$ absolute and $1.33\times$ relative differences in medians of schools which do and do not complete a year, $p = 0.865$), higher education participation rate ($+2\%$ and $1.04\times$, $p = 0.677$), indices of multiple deprivation ($-4.30$ and $0.86\times$, $p = 0.219$), or ethnic diversity ($-3\%$ and $0.93\times$, $p = 0.844$). We do not perform this test for gender variation since the data is less reliable, as previously mentioned. Therefore, the typical societal barriers to STEM do not appear to affect schools' ability to succeed within the PRiSE framework.

We note that as well as schools dropping out, even within those schools which complete a year there is typically some reduction in the number of students that persist with project work. As our reporting only recorded total numbers of students by Key Stage (see Table B1 for more information) at events rather than on a school-by-school basis, we cannot calculate student retention rates for schools which received on campus kick-off events, since these involved multiple schools some of which subsequently dropped out. Neglecting those schools, we find that between 2015–2019 the overall retention rate from kick-off to conference was $56 \pm 3\%$ ($n = 321$), though we note individual schools' rates varied widely with an interquartile range of

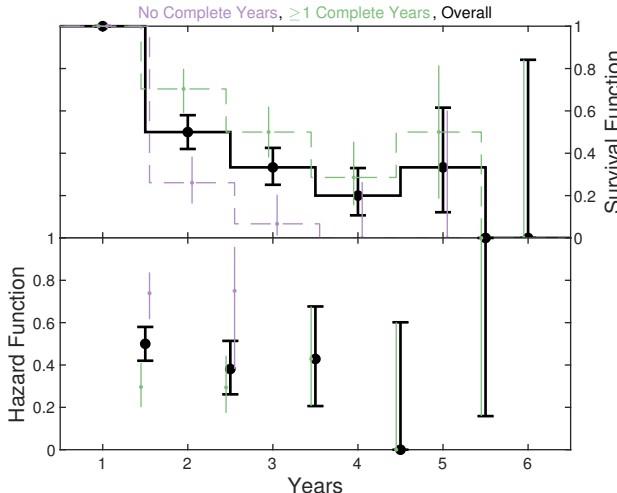

**Figure 5.** Schools' survival (top) and hazard (bottom) functions across years in a similar format to Figure 4. Overall results in black, with colours subdividing by schools which did (green) and did not (purple) complete at least one year.

39–88% and 7 schools (out of the 27 considered) retained all students throughout the programme. Similar to with participation, future work (subject to funding and ethics approval) could investigate the retention within PRiSE at the student-level and whether this is also equitable.

### 3.2 Across years

We have seen considerable repeated buy-in from many schools over multiple years with PRiSE (see Appendix A for the data), thus we also investigate retention across years. Figure 5 shows the overall results in black where only schools which began work on projects are included. We note that because the programme has been carefully grown since its inception, not all schools started at the same time and this is why the survival function is not strictly decreasing, e.g. only 6 schools could have been involved for 5 years and just one for the full 6 years. Overall the drop-off rate is consistent year-on-year at $45 \pm 10\% \, \text{year}^{-1}$ (averaged over the first 2 years for which we have better statistics). Similarly to before, there is unfortunately no suitable benchmark for comparison since no similar programmes have yet reported on university–school relationships built over several years of running the same programme. Again we find no significant differences in schools which participate in PRiSE for multiple years compared to those which don't in terms of school category ($\chi^2(4) = 2.22$, $p = 0.695$), free schools meals percentage ($+5\%$ absolute and $1.43\times$ relative, $p = 0.327$), higher education participation rate ($-1\%$ and $0.97\times$, $p = 0.458$), indices of multiple deprivation ($+0.75$ and $1.03\times$, $p = 0.371$) or ethnic diversity ($+12\%$ and $1.32\times$, $p = 0.111$). However, as shown in Figure 5, schools which have been able to successfully complete at least one year (green) are far more likely to participate again at $70 \pm 10\%$ compared to those which have not (purple) at $26 \pm 11\%$. Importantly though, that latter value is not negligible showing that some schools are willing to try again. So far (not including academic year 2019/20) there have

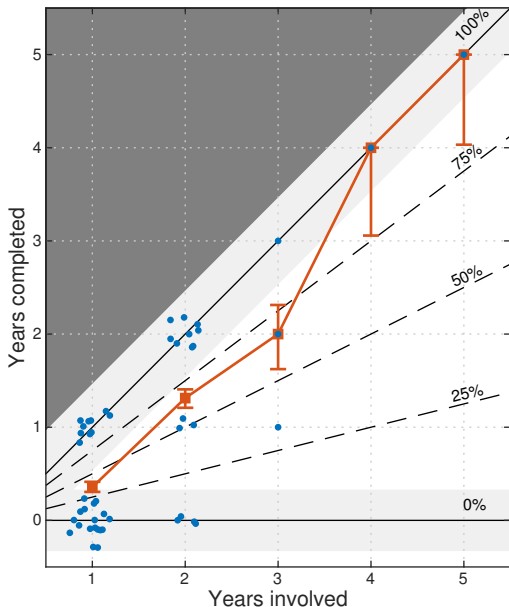

**Figure 6.** The number of years each school completed a PRiSE project against the number of years they were involved. Datapoints (blue) have been jittered for visibility. The average number of years completed and standard Clopper and Pearson (1934) interval for this rate are also shown (red).

been three out of a potential seven second attempts where this has led to subsequent success, with these largely being run with the same teacher rather than a different one.

The link between the success of schools at completing projects and the continued involvement of schools across several years is highlighted further in Figure 6. This shows for each school listed in Appendix A the number of years that they completed
PRiSE projects against the total number of years they started them. Note that we do not include academic year 2019/20 in either dimension due to disruption by the COVID-19 pandemic. It appears that schools predominantly tend to lie near the two possible extremes (solid lines) of either completing projects every year they were involved or not completing the projects at all. The average rate and its uncertainty (constructed by treating each school as a single binomial experiment and combining them using Bayes' theorem) are also shown in the figure in red as a function of the number of years involved. A clear trend can
be seen that completion rates increased as schools were involved for more years. However, one must be careful in interpreting possible causation behind this correlation. While there are examples where a second attempt at a PRiSE project did lead to improved retention, in most cases it is likely that schools which succeed are simply more likely to participate again whereas those that (for whatever reason) don't succeed are understandably less likely to continue in subsequent years.

While at our conferences teachers are unanimous in their intentions to participate again when asked via paper questionnaire,
the data in Figure 5 show a fairly consistent drop-off rate of $30 \pm 12\%\,\mathrm{year}^{-1}$ from schools which completed a year. The majority of this might be explained by teacher turnover, which in 2015 stood at $21\%$ per year nationally ($13\%$ left teaching altogether) with an increasing year-on-year trend and London having been highlighted as having a much higher churn rate

(Worth et al., 2017). We are aware (often through out of office messages or bounced emails) of numerous teachers having either moved schools or left the profession and it is rare that they communicate this with us ahead of time, handover project responsibilities to another teacher, or take projects with them to their new school. Establishing the exact number of schools which did not return due to this reason compared to other explanations would require considerable resources beyond the scope of this evaluation though. Teacher turnover poses a real challenge to establishing lasting relationships with schools, when these relationships are so dependent on individuals and are often not embedded within the schools themselves — though the same may also be said of universities, e.g. most of the public engagement professionals who led the Beacons programmes in the UK (National Coordinating Centre for Public Engagement, 2008) don't work in public engagement anymore.

Maximising the retention of schools across years is necessarily a function of the capacity of a programme. While bringing new schools into the programme is certainly beneficial, we have seen that teachers' ability and confidence in supporting project work in their school develops not only across the 6-month programme but over several years. Therefore, a practical balance could be to aim to involve schools directly for just a few years to the point that they can sustainably run projects with fewer interventions from the university, perhaps just an on campus kick-off and then the conference as researcher capacity is less of an issue with these. We acknowledge though that some schools might no longer participate without the full suite of interventions. This approach would enable the wider impacts on schools that benefit from multiple years of participation while also ultimately freeing up capacity in the long-term for new schools to be able to benefit from the programme.

## 4  Feedback

Here we present qualitative data from teachers relating to the issues of accessibility, diversity, equity, and retention.

### 4.1  Method

We prefer to take a holistic approach in investigating teachers' thoughts about these issues, using evaluative data from a variety of methods throughout the programme. Formal feedback from teachers has been gathered from teachers via paper questionnaires handed out at our student conferences each year (in 2020 due to the COVID-19 pandemic an online form was instead used). All data used here were in response to open questions and further details of this questionnaire can be found in Archer et al. (2020). In addition to this formal feedback, we also use data obtained more informally. These include comments made by teachers in-person throughout the programme, e.g. during researcher visits or our conference, and those passed on via email. Where possible explicit consent has been obtained to use these comments, though in general by participating in PRiSE teachers are aware that they are entering a relationship of mutual trust and that any information passed on by them will be used with integrity (cf. BERA, 2018). These informal comments are recorded at the time and then analysed along with and in the same way as the formal feedback. The anonymity of the teachers and their schools are protected, quoting only the school's pseudonym as well as the project and year the feedback related to along with the method the data was obtained.

All qualitative data were analysed using thematic analysis (Braun and Clarke, 2006), with the themes being allowed to emerge from the data via grounded theory (Robson, 2011; Silverman, 2010) as follows:

1. Familiarisation: Responses are read and initial thoughts noted.

2. Induction: Initial codes are generated based on review of the data.

3. Thematic Review: Codes are used to generate themes and identify associated data.

4. Application: Codes are reviewed through application to the full data set.

5. Analysis: Thematic overview of the data is confirmed, with examples chosen from the data to illustrate the themes.

In the following subsection, we highlight the different themes identified relating to aspects of accessibility, diversity, equity, and retention through bold text, providing illustrative quotes.

## 4.2 Results and discussion

Several teachers from a variety of different schools have raised that they **value the diversity** present across PRiSE, particularly that students from different schools and backgrounds are able to interact as equals at the student conferences.

"*They love the competition with independent schools*" (Teacher, Rushmore Academy, MUSICS 2018, in-person)
"*Giving the students the opportunity to meet with other schools and academic staff in person is a major highlight*" (Teacher, Tree Hill High School, PHwP 2020, questionnaire)

One teacher expanded on this, contrasting the diversity present to other schemes and positing that the support provided through PRiSE enabled this difference

"*At* [PRiSE student conference] *Cosmic Con, pupils from a diverse range of state and independent schools have the opportunity to share their experiences and discuss their findings with each other, widening their perspective from the natural micro-habitat of the school environment to the wider community around them. Groups that have worked on the same project naturally gravitate into discussion. The key to increasing collaboration and empathy for those in different schools is to put pupils in the same place, at the same time, with some interest or experience in common to talk about. There are other schools conferences around, but QMUL has such close links with and provides such support for schools, that this adds to the diversity of Cosmic Con.*" (Teacher, Octavian Country Day School, PHwP 2019, email testimonial following in-person comment)

The equality of all students at PRiSE conferences is further reflected in that prizes awarded to students, judged by researchers, have been well distributed amongst different types of schools with no obvious biases (though due to the small numbers we do not perform a thorough statistical analysis). This has, however, had a prodigious effect on students from disadvantaged backgrounds

"*The students were buzzing on their journey back, they kept saying 'I can't believe our little comprehensive won'.*" (Teacher, Coal Hill School, MUSICS 2016, in-person)

Teachers' comments have highlighted that while some PRiSE projects are thought to be equitable in terms of **students' ability**, that unfortunately may not currently the case with all of them

> "*The initial information received on the project was quite daunting. But the presentation introduction to it was much more accessible. The scope of the project was accessible to students of all abilities.*" (Teacher, Spence Academy for Young Ladies, MUSICS 2018, questionnaire)

> "*It is really appropriate in content and context for my students... Throughout the project there have been tasks that can meet the varying abilities of my students. Really well designed project.*" (Teacher, Bending State College, PHwP 2020, questionnaire)

> "*It gives students a chance to do something very interesting, if they are enthusiastic. There isn't much to do for students who are struggling. The projects are very high demand, both in time and skill. This has always been our problem with retaining students' interest over several months. I'm not sure how it can be changed too much without making it boring for students, it's a difficult balance. I think just a variety in skill level, it's nice to have a whole class working on a project together because it pulls everyone up, so there should be stuff for the E/D-grade students to have a go at as well as the A/A\* pupils. Some projects don't lend themselves very well to that, but others like MUSICS do because anyone can listen to some sounds but there was still stuff for the higher skill students to get stuck into.*" (Teacher, Hill Valley High School, ATLAS 2020, questionnaire)

In addition to simply tweaking the projects to reduce their barrier to entry while still maintaining their broad open-ended scope, there are other potential ways to address this also. One teacher (Xavier's Institute for Higher Learning, MUSICS 2017) highlighted in-person that the students' group dynamics can play a big role in their successful participation. In the previous year a clear enthusiastic leader had emerged who could include everyone in the project in different ways appropriate to their ability, whereas such leadership had not successfully been established amongst any of the students the following year leading to the group struggling to find a direction and cultivate eagerness in everyone participating until the researcher visit occurred. We suggest that teachers might be able to help facilitate establishing the group dynamics where required, since they are more familiar with the students. Another possibility is peer mentoring from the school's previous year of PRiSE students.

> "*I have had a lot of help from Year 13 students acting as mentors... this has been a useful exercise in peer learning.*" (Teacher, Xavier's Institute for Higher Learning, MUSICS 2017, questionnaire)

Of course this is only a viable option for schools which complete one year in the first instance. Some teachers who have expressed interest in capitalising on peer mentoring have often struggled to implement it within their schools though. Furthermore, some schools prefer to change project after a couple of years and so the ability of previous PRiSE students to effectively mentor, apart from in a more pastoral capacity, may be limited. While further research is required into what makes for successful group dynamics or student mentorship, we nonetheless hope to be able to include something on both of these aspects in planned 'how to' guides for teachers.

Finally, we discuss the issue of retention within the programme. Unfortunately, given the often poor **communication** from teachers (also highlighted as an issue within the ORBYTS programme, Sousa-Silva et al., 2018) it is not always clear as to

why individual schools drop out. We have little evidence around why teachers do not communicate this, though one teacher (Colonial Fleet Academy, MUSICS 2016, email) who had been unresponsive eventually expressed a feeling of embarrassment that all their students bar one had ceased project work. While we try and assure teachers upon initiating the projects that we anticipate some drop-off and that there is no pressure from us to remain involved, more may be required in this area. From the minority of teachers that do inform us of their school dropping out, typically via email, reasons have included mock exams getting in the way, students losing interest, realising the amount of work involved, difficulty balancing the project with other activities / their normal school workload, giving up due to uncertainty in how to progress, and not feeling like they've made enough progress to continue. Similar themes have been expressed from teachers in-person concerning some (but not all) of their students dropping out. We feel that many of these issues might have been mitigated through more and/or earlier communication from the school to us, as we have been able to assist with similar struggles at more communicative schools. Despite a clear support process being laid out at kick-off workshops a number of teachers from schools that successfully completed projects realised in hindsight that they should have taken advantage of the opportunities from the university earlier in the programme than they did

> "*Should have taken advantage* [of support from Queen Mary] *at* [an] *earlier stage.*" (Teacher, St Trinians, SCREAM 2018, questionnaire)
>
> "*The call with Dr Senz would have been more helpful earlier on - it made a big difference, and students would have got more out of the project if they had more time after this took place.*" (Teacher, Imperial Academy, ATLAS 2020, questionnaire)

This is again something that could be stressed in teacher guidance upon engaging with the programme to better set expectations and good practice, as many new teachers to the programme may not be used to such a reactive way of working rather than the typical 'push' model from teachers to students.

Overall, we get the impression that retention within PRiSE can often come down to the individual teacher – those that are communicative and properly engage with the programme and its expectations from the outset are far more likely to see their students succeed. This is backed up by evidence from several schools, where a change of teacher has either led to increased engagement with the programme (e.g. Rydell High School), success at previously unsuccessful schools (e.g. Hill Valley High School), or unfortunately previously successful schools dropping out of the programme (e.g. Prufrock Preparatory School). One teacher who changed schools (from Hogwarts to Prufrock Preparatory School) bringing PRiSE projects with them also raised with us in-person that the culture within the schools can play a role in students' engagement with the programme (and extra-curricular activities in general). Both of these are challenging issues to address as we aim to increase the retention and equity of the programme. While it is clear that more detailed qualitative research is required in this area, perhaps giving clearer information and expectations upon signup as well as providing further guidance on ways of successfully integrating and nurturing project work within schools, as highlighted by other teachers, these issues might be somewhat mitigated.

# 5 Conclusions

Societal inequalities in access to and engagement with science are prevalent even in secondary/high schools (Hamlyn et al., 2020). While university engagement programmes, like independent research projects, could address these issues, at present few such schemes specifically target traditionally under-represented groups and in general globally students' participation in such projects are inequitable (Bennett et al., 2016, 2018). In this paper we have evaluated the accessibility, diversity, and equity of the 'Physics Research in School Environments' (PRiSE) programme of independent research projects (Archer et al., 2020).

The schools involved in PRiSE have been benchmarked against those participating in similar programmes of research-based physics projects for schools in the UK. Investigating measures of the socio-economic status, race, and genders of the schools' students have revealed that PRiSE has engaged much more diverse groups of schools with substantially more under-represented groups than is typical. Indeed, PRiSE schools are mostly reflective of national statistics and in some measures feature an over-representation of disadvantaged groups. While PRiSE has featured fewer independent and selective schools than other schemes, the proportions are not currently reflective of all schools nationally and thus new policies have been implemented to improve diversity in these regards.

Survival analysis has been used to explore the retention of schools within the programme. This was firstly done across the different intervention stages of PRiSE within each academic year. We find a fairly consistent drop-off rate throughout, with no significant differences between the different projects or the years considered. While little research into the retention of schools within protracted programmes of engagement currently exist, the rates exhibited by PRiSE are at least similar to another programme (Hope-Stone Research, 2018). The analysis has highlighted the importance of PRiSE's researcher-involvement in the schools' success. This is despite independent research projects in general often not being supported by external mentors (Bennett et al., 2016, 2018). No biases in schools' retention appear present by school category, socio-economic background, or race. This suggests that schools' ability to succeed at independent research projects is independent of background within the PRiSE framework. PRiSE has seen repeated buy-in over multiple years from numerous schools. Hence we also looked at the retention of schools across multiple years, again finding no real differences in the backgrounds of schools which return and those which do not. Indeed, the only predictor for multiple years of participation is whether the school engaged with the programme through to completion for at least one year. Our interpretation is that success within PRiSE often comes down to the individual teacher, with poor communication (cf. Sousa-Silva et al., 2018) or not fully engaging with the programme and its expectations serving as key barriers in schools' participation.

Qualitative feedback from teachers have shown that they value the diversity within the programme, seeing the ability of students from different schools and backgrounds to interact as equals at PRiSE conferences as a positive aspect. They also attribute this equity to the exceptionally high level of support provided by PRiSE to the schools involved. The need for slight modifications to make some of the projects more accessible to students of all abilities has been raised. These concerns might also be addressed by prompting teachers to facilitate students' group dynamics and potentially incorporating peer mentoring from previous years' PRiSE students, both of which have been reported as successful in some cases but with mixed results at other schools. More teacher guidance, co-created with teachers themselves, on the expectations within the programme as well

as good practice in incorporating and nurturing project work within schools could be provided to help with retention in new schools.

Our analysis has been limited to the London geographic area, so it is not yet clear that the PRiSE framework of independent research projects would necessarily be as accessible or equitable in different parts of the UK or in different countries. With the adoption of this approach to engagement at other universities, however, we hope to be able to investigate this in the future. Further, only school-level metrics have been considered here and more detailed analysis at the individual student level and their characteristics could be considered in future, which would require funding to commission such research by social scientists along with the necessary ethical approval. Finally, in-depth qualitative research into the reasons behind schools dropping out of PRiSE, both within the six-month programme and between different years, would be beneficial in understanding what the current barriers to prolonged participation are and how these could be best addressed in the future.

## Appendix A: PRiSE schools

Below is a table with information about all the schools which have been involved in PRiSE. To protect the anonymity of students and teachers, pseudonyms (taken from https://annex.fandom.com/wiki/List_of_fictional_schools) have been used. The table details the year the school joined, how many years they have undertaken projects, the number of these years they successfully completed (i.e. made it all the way to the student conference), along with categorical information, and deciles (used here to further protect anonymity) across their catchment areas of the schools' percentage of students on free school meals (FSM), higher education participation rate (HEPR), indices of multiple deprivation score (IMD), and percentage of ethnic minorities (EM). Further information about these are given in Appendices B and C. Missing data in the table is due to it not being publicly available. Different schools partnerships where schools have worked together (or have at least attempted to) are indicated by letters. Schools which signed up for but never commenced project work (by hosting/attending a kick-off meeting) are not included here. Note that the years completed column does not include data from the 2019/20 academic year due to disruption by the COVID-19 pandemic and we mark all schools affected by this with an asterisk (*) in this column.

| Joined | School pseudonym | Category | Admissions | Gender | FSM | HEPR | IMD | EM | Years | Completed | Partnership |
|---|---|---|---|---|---|---|---|---|---|---|---|
| 2014 | Hogwarts | Independent | | Mixed | | | | | 5 | 5 | a |
| 2015 | Colonial Fleet Academy | Academy | Non-Selective | Mixed | 7 | 2 | 8 | 8 | 1 | 1 | |
| 2015 | Constance Billard School for Girls | LA maintained | Non-Selective | Girls | 8 | 6 | 9 | 10 | 2 | 0 | b |
| 2015 | St. Judes School for Boys | LA maintained | Non-Selective | Boys | 9 | 6 | 9 | 10 | 2 | 0 | b |
| 2015 | Sweet Valley High School | Academy | Non-Selective | Mixed | 5 | 9 | 7 | 10 | 2 | 2 | |
| 2015 | Xavier's Institute for Higher Learning | College | | Mixed | | | | | 5 | 4* | |
| 2016 | Angel Grove High School | LA maintained | Non-Selective | Girls | 6 | 7 | 7 | 10 | 1 | 1 | c |
| 2016 | Coal Hill School | LA maintained | Non-Selective | Girls | 6 | 6 | 7 | 9 | 2 | 2 | |
| 2016 | Earth Force Academy | Academy | Non-Selective | Mixed | 3 | 8 | 4 | 8 | 1 | 1 | |
| 2016 | Hill Valley High School | LA maintained | Non-Selective | Mixed | 8 | 6 | 9 | 10 | 3 | 1* | d |
| 2016 | Hillside Academy | Academy | | Mixed | | | | | 1 | 0 | |
| 2016 | Imperial Academy | Academy | Selective | Boys | 1 | 9 | 4 | 9 | 3 | 2* | e |
| 2016 | Our Lady of Perpetual Sorrow | Independent | | Girls | | | | | 1 | 0 | |
| 2016 | Prufrock Preparatory School | Independent | | Mixed | | | | | 3 | 2 | a |
| 2016 | Springfield Community College | College | | Mixed | | | | | 1 | 1 | |

| Joined | School pseudonym | Category | Admissions | Gender | FSM | HEPR | IMD | EM | Years | Completed | Partnership |
|---|---|---|---|---|---|---|---|---|---|---|---|
| 2016 | St Trinians | Independent | | Girls | | | | | 4 | 3* | a |
| 2016 | Stone Canyon High School | LA maintained | Non-Selective | Boys | 7 | 7 | 8 | 10 | 1 | 1 | c |
| 2016 | Stoneybrook Academy | Academy | | Mixed | | | | | 2 | 0 | |
| 2016 | Tree Hill High School | LA maintained | Non-Selective | Mixed | 7 | 9 | 6 | 10 | 4 | 1* | d |
| 2016 | Vulcan Science Academy | Academy | Non-Selective | Mixed | 6 | 10 | 6 | 9 | 1 | 0 | a |
| 2017 | Avalanche Arts Academy | Academy | Non-Selective | Mixed | 3 | | | | 1 | 0 | |
| 2017 | Barcliff Academy | Academy | Non-Selective | Mixed | 5 | 8 | 8 | 10 | 1 | 0 | |
| 2017 | Boston Bay College | College | | Mixed | | | | | 3 | 2* | |
| 2017 | Bronto Crane Academy | Academy | Non-Selective | Mixed | 8 | 8 | 7 | 10 | 2 | 1 | d |
| 2017 | Chalet School | Independent | | Boys | | | | | 1 | 0 | |
| 2017 | Fire Nation Academy for Girls | LA maintained | Non-Selective | Girls | 8 | 8 | 8 | 9 | 1 | 0 | f |
| 2017 | Harbor School | Independent | | Mixed | | | | | 3 | 2* | f |
| 2017 | Io House | Independent | | Mixed | | | | | 1 | 1 | |
| 2017 | Kelsey Grammar School | Academy | Selective | Mixed | 1 | | | | 1 | 0 | |
| 2017 | Martha Graham Academy | Academy | Non-Selective | Mixed | 5 | 8 | 8 | 9 | 3 | 0* | f |
| 2017 | Miss Shannon's School for Girls | LA maintained | Non-Selective | Girls | 4 | 8 | 6 | 9 | 1 | 0 | |
| 2017 | Roosevelt High | LA maintained | Non-Selective | Mixed | 8 | 4 | 9 | 9 | 2 | 1 | |
| 2017 | Rydell High School | Free School | Non-Selective | Mixed | 1 | 10 | 5 | 10 | 3 | 2* | |
| 2017 | Smeltings | Independent | | Boys | | | | | 2 | 2 | |
| 2017 | Spence Academy for Young Ladies | Academy | Selective | Girls | 1 | 8 | 4 | 8 | 3 | 2* | |
| 2017 | St. Francis Academy High School | Academy | Non-Selective | Mixed | 7 | 10 | 5 | 9 | 1 | 0 | |
| 2017 | Stoneybrook Day School | Independent | | Mixed | | | | | 1 | 1 | |
| 2017 | Sunnydale High School | Academy | Non-Selective | Mixed | 9 | 8 | 9 | 9 | 3 | 2* | |
| 2017 | Washington Preparatory Academy | Independent | | Mixed | | | | | 1 | 0 | |
| 2018 | American Eagle Christian School | LA maintained | Non-Selective | Mixed | 3 | 7 | 7 | 10 | 1 | 0 | |
| 2018 | Bel-Air Academy | Academy | Non-Selective | Mixed | 7 | 8 | 8 | 10 | 2 | 0* | |
| 2018 | Bending State College | College | | Mixed | | | | | 1 | 0* | |
| 2018 | Octavian Country Day School | Independent | | Mixed | | | | | 2 | 1* | |
| 2018 | Pokémon Technical Institute | LA maintained | Non-Selective | Mixed | 9 | 7 | 9 | 10 | 2 | 1* | |
| 2018 | Prescott Academy for the Gifted | LA maintained | Selective | Boys | 3 | 9 | 6 | 10 | 1 | 1 | |
| 2018 | Rushmore Academy | Academy | Non-Selective | Mixed | 8 | 10 | 6 | 9 | 1 | 0 | |
| 2018 | Summer Heights High | LA maintained | Non-Selective | Mixed | 5 | 7 | 8 | 10 | 1 | 0 | |
| 2018 | Thomas Aquinas Private Girls' School | Independent | | Girls | | | | | 1 | 0 | |
| 2018 | Walford High School | Free School | | Mixed | 1 | | | | 2 | 0* | |
| 2018 | Worcestershire Academy | Academy | Non-Selective | Mixed | 8 | 7 | 9 | 10 | 1 | 0 | |
| 2019 | Bullworth Academy | Academy | Non-Selective | Mixed | 10 | 8 | 9 | 10 | 1 | * | |
| 2019 | Chuck Norris Grammar School | Academy | Non-Selective | Boys | 5 | 7 | 8 | 9 | 1 | * | |
| 2019 | Holy Forest Academy | Academy | Non-Selective | Mixed | 7 | 6 | 7 | 10 | 1 | * | |
| 2019 | Jedi Academy | Academy | Selective | Girls | 2 | 8 | 5 | 9 | 1 | * | e |
| 2019 | Marlin Academy | Academy | Non-Selective | Boys | 6 | | | | 1 | * | |
| 2019 | Morningwood Academy | Academy | Non-Selective | Mixed | 2 | 7 | 4 | 9 | 1 | * | |
| 2019 | Orbit High School | Free School | | Mixed | 1 | | | | 1 | * | g |
| 2019 | Quirm College for Young Ladies | Independent | | Girls | | | | | 1 | * | h |
| 2019 | Royal Dominion College | Independent | Selective | Mixed | | | | | 1 | * | |
| 2019 | Sky High | Free School | | Mixed | 5 | | | | 1 | * | h |
| 2019 | Smallville High School | Free School | Selective | Mixed | | | | | 1 | * | |
| 2019 | Starfleet Academy | Academy | Non-Selective | Mixed | 9 | 8 | 9 | 10 | 1 | * | |
| 2019 | Stoolbend High School | Special school | | Mixed | 10 | | | | 1 | * | |
| 2019 | Summer Bay High | LA maintained | Non-Selective | Mixed | 7 | 8 | 8 | 10 | 1 | * | |
| 2019 | Sycamore Secondary School | Free School | Non-Selective | Mixed | 8 | 3 | 8 | 9 | 1 | * | h |
| 2019 | Warren Greeley Preparatory School | Independent | | Mixed | | | | | 1 | * | g |

| Key Stage | Year | Final Exam | Age | Policy |
|-----------|------|-----------|-----|--------|
| KS4 | 10 | None | 14–15 | Compulsory |
| | 11 | GCSE | 15–16 | |
| KS5 | 12 | AS-Level (optional) | 16–17 | Optional |
| | 13 | A-Level | 17–18 | |

**Table B1.** Summary of the stages in the English education system applicable to PRiSE.

| Joined | School pseudonym | Category | Admissions | Gender | FSM | HEPR | IMD | EM | Years | Completed | Partnership |
|--------|-----------------|----------|-----------|--------|-----|------|-----|----|----|-----------|-------------|
| 2019 | Welton Academy | Academy | Non-Selective | Mixed | 4 | 10 | 3 | 8 | 1 | * | h |

**Appendix B:  Information about UK/English schools**

This paper uses the context of the UK/English education system to assess the diversity of schools engaged within the PRiSE (and other) programme(s). To those unfamiliar with this system, we provide some further notes here. Schools are classified by the Department for Education (2018) into the following main categories:

– **Academies** are schools that are state funded and free to students but are not run by the local authority. They have much more independence than most other schools including the power to direct their own curriculum. Academies are established by sponsors from business, faith or voluntary groups in partnership with the Department for Education working with the community.

– **Colleges** are post-16 education establishments not part of a secondary school.

– **Free Schools** are a type of academy set up by teachers, parents, existing schools, educational charities, universities, or community groups.

– **Independent Schools** are funded by the fees paid by the parents of pupils, contributions from supporting bodies and investments. They are not funded or run by central government or a Local Authority. They can set their own curriculum.

– **Local Authority (LA) Maintained Schools** are wholly owned and maintained by Local Authorities and follow the
470 national curriculum.

Other less common categories are not considered in this paper due to small number statistics. In addition to the school category, UK schools can also be classified by their admissions policy. Selective (or grammar) schools enrol pupils based on ability whereas non-selective (or comprehensive) schools are not able to do this. While some independent schools are selective, not all of them are. Table B1 shows how school years are denoted in the English system, with other contextual information.

## Appendix C: Method for gathering metrics on UK schools

In this paper we look at several metrics to assess the backgrounds of the schools' pupils. School categories are listed in Edubase (Department for Education, 2018), as are the percentage of students on free school meals (though this is typically not listed for colleges, independent schools, and some academies). Other relevant metrics are not included in Edubase but are tied to census lower and middle layer super output areas (LSOAs and MSOAs respectively, Office of National Statistics, 2011). However, since schools will draw students from a wider range of locations than simply the census area within which they are located, this ideally necessitates knowledge of a school's catchment area to gain a better understanding of the backgrounds of the schools' students. While such information is available for state schools in the Greater London area through the London Schools Atlas (Greater London Authority, 2014), which lists for each school the resident LSOAs/MSOAs of their student base, unfortunately there is no equivalent publicly accessible data covering the entire UK.

Here we detail how school metrics across their entire catchment areas are calculated. Higher education participation rates for each school are determined using the number of entrants to higher education and cohort populations across each school's MSOAs from POLAR4 data (Office for Students, 2018). Similarly, index of multiple deprivation scores (Ministry of Housing, Communities & Local Government, 2015) are averaged over each school's LSOAs. For protected characteristics such as gender/sex and race/ethnicity, we do not collect this data from students for ethics reasons. In the latter case, while it has been observed by session leaders at interventions that a diversity of ethnicities have been involved, we opt to quantify this through the ethnic diversity of the areas from which students are drawn. Census data (Office of National Statistics, 2011) on ethnic groups is used to calculate the percentage of people from ethnic minorities (i.e. non-white groups) across each school's LSOAs.

To still enable some comparison between PRiSE and the national programmes, for which we do not have access to information on schools' catchment areas, we rely on using the metrics pertaining only to the LSOA/MSOA within which the school resides. These results are indicated by dashed lines in Figure 3 and have also been computed for PRiSE schools to ensure like-for-like comparisons. Across the Greater London area we can check the reliability of these local proxies and comparisons are shown in Figure C1. These reveal that the local and full catchment distributions appear similar, with location and scale parameters (e.g. means and standard deviations respectively) that differ only by a few percent / score points. Therefore, taking into account schools' full catchment area only slightly changes the underlying distributions of the societal measures, apart from in the extremes of the distributions (i.e. the tails) where greater differences occur. However, while the local census and full catchment data for London schools correlate, this correlation is not particularly strong (the correlation coefficients are $R = 0.64$, $0.65$, and $0.85$ respectively) and the linear best fit lines have slopes significantly less than unity. This highlights that the metrics vary substantially across all the census areas a school draws students from, meaning that a school can have rather different values when either using local or full catchment data. Further investigation of these societal measures applied to schools in general is beyond the scope of this paper.

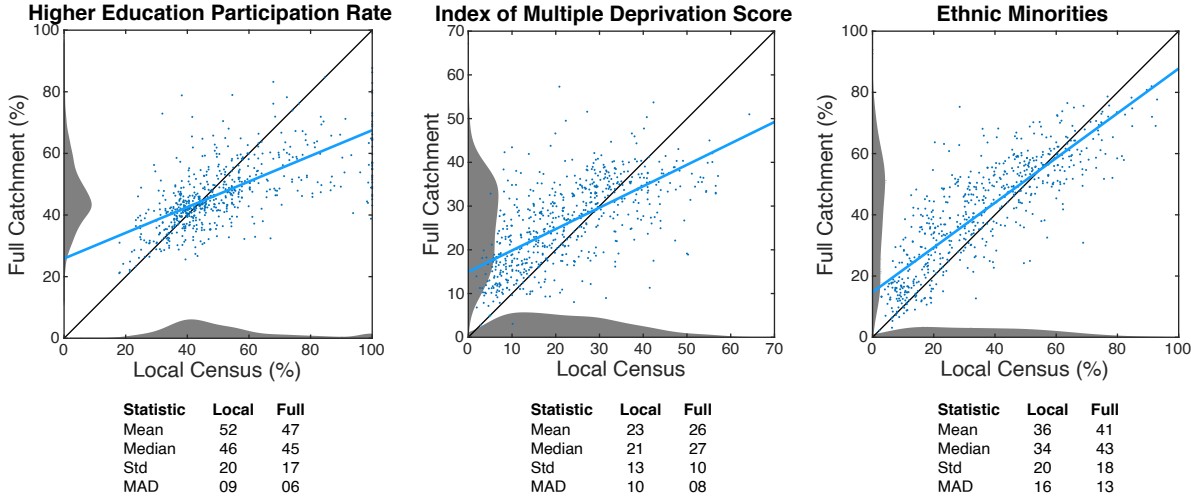

| Statistic | Local | Full |
|-----------|-------|------|
| Mean | 52 | 47 |
| Median | 46 | 45 |
| Std | 20 | 17 |
| MAD | 09 | 06 |

| Statistic | Local | Full |
|-----------|-------|------|
| Mean | 23 | 26 |
| Median | 21 | 27 |
| Std | 13 | 10 |
| MAD | 10 | 08 |

| Statistic | Local | Full |
|-----------|-------|------|
| Mean | 36 | 41 |
| Median | 34 | 43 |
| Std | 20 | 18 |
| MAD | 16 | 13 |

**Figure C1.** Comparison of local census and full catchment data for all schools across Greater London ($n = 625$), with a linear regression (blue line) and marginal distributions (grey areas) shown. Location (mean and median) and scale (standard deviation and median absolute deviation) parameters for the two types of data are also displayed.

**Appendix D: Retention data**

The below table contains data on the retention of schools within PRiSE across intervention stages. We use the following terminology:

– **Attended**: schools which received the intervention

– **Didn't attend**: schools which did not receive the intervention but were still engaged with the programme at that stage

– **Unresponsive**: schools that did not respond to our communications from that point on

– **Dropped out**: schools that communicated their dropping out the programme at that stage

Schools that have become unresponsive or drop out are no longer counted in the table for subsequent intervention stages.

| | | 2017/18 | | | | 2018/19 | | | |
|---|---|---|---|---|---|---|---|---|---|
| | | SCREAM | MUSICS | PHwP | ATLAS | SCREAM | MUSICS | PHwP | ATLAS |
| Assignment | Assigned | 5 | 18 | 7 | 7 | 4 | 24 | 8 | 8 |
| Kick-off | Attended | 3 | 14 | 4 | 4 | 4 | 13 | 5 | 3 |
| | Didn't attend | 0 | 0 | 1 | 1 | 0 | 1 | 0 | 1 |
| | Unresponsive | 1 | 1 | 0 | 2 | 0 | 6 | 3 | 3 |
| | Dropped out | 1 | 2 | 2 | 0 | 0 | 4 | 0 | 1 |
| Visit | Attended | 3 | 7 | 1 | 1 | 4 | 5 | 4 | 0 |
| | Didn't attend | 0 | 4 | 2 | 2 | 0 | 4 | 1 | 3 |
| | Unresponsive | 0 | 2 | 1 | 1 | 0 | 5 | 0 | 1 |
| | Dropped out | 0 | 1 | 1 | 1 | 0 | 0 | 0 | 0 |
| Comments | Attended | 3 | 5 | 0 | 1 | 2 | 3 | 2 | 1 |
| | Didn't attend | 0 | 3 | 3 | 1 | 2 | 1 | 3 | 2 |
| | Unresponsive | 0 | 2 | 0 | 0 | 0 | 3 | 0 | 0 |
| | Dropped out | 0 | 1 | 0 | 1 | 0 | 2 | 0 | 0 |
| Conference | Attended | 3 | 8 | 3 | 2 | 4 | 4 | 4 | 3 |
| | Dropped out | 0 | 0 | 0 | 0 | 0 | 0 | 1 | 0 |

*Data availability.* Data supporting the findings of this study that is not already contained within the article or derived from listed public domain resources are available on request from the corresponding author. This data is not publicly available due to ethical restrictions based on the nature of this work.

*Author contributions.* MOA conceived the programme and its evaluation, performed the analysis, and wrote the paper.

*Competing interests.* The author declares that they have no conflict of interest.

*Acknowledgements.* We thank Dominic Galliano, Olivia Keenan, Charlotte Thorley, and Jennifer DeWitt for helpful discussions. MOA holds a UKRI (STFC / EPSRC) Stephen Hawking Fellowship EP/T01735X/1 and received funding from the Ogden Trust. This programme has been supported by a QMUL Centre for Public Engagement Large Award, and STFC Public Engagement Small Award ST/N005457/1.

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
