# Peer review of "Schools of all backgrounds can do physics research: On the accessibility and equity of the PRiSE approach to independent research projects"

_Geoscience Communication, 2020_

## Short Comment (SC1) · 8 Aug 2020

One thing I have noticed is principle investigators on research projects using citizen scientists as cheap labor. By that I mean they advertise for people to contribute and then instruct them to do menial tasks such as data transcription and collation, especially in climate science. IMO, this is the last thing that you want students to be doing – expect much greater things from them. Let them work the algorithms and mathematical physics and encourage them to find the next great ansatz that might lead to a research breakthrough. That's all I have to say, because if history is any indication, insight can

come from anywhere.

---

## Referee Comment (RC1) · Michael Reiss (Referee) · 19 Aug 2020

This is a valuable and well-written submission. It tackles an important issue and makes good links with the existing literature; the analysis is excellent and the findings add considerably to what is already known in the published literature. There is a degree of self-congratulation in the comparisons with other programmes – but the comparisons are very interesting!

1. I have one major comment. It is a huge pity that "for ethical reasons we did not collect

any protected characteristics (such as gender or race) or sensitive information (such as socio-economic background) from the students involved" (lines 67-68). Such data are not infrequently collected by educational researchers (indeed, they are collected by the DfE and available in the NPD) and I note the paragraph on gender that spans pages 7 and 8 (some might object to identifying gender in this way, though I am less of a purist). As the author is well aware, this means that all the conclusions made can only be made at school rather than individual student level. This, I am afraid, is not a trivial point. It is perfectly possible that the students who participate in these projects are far from representative of their schools. I think this should be made much clearer in the submission – in my view even the "School students from all backgrounds can do physics research" in the title is misleading and needs changed.

2. I suspect the "issue" with IRIS is not in "their targeting" (line 155) but which schools respond to its offer.

3. I think it would be worth discussing briefly whether maximising retention of schools across years is always a good.

4. Was there any ethical clearance for the research element of the work?

5. With reference to the qualitative data, there is a clear account of thematic analysis but then no evidence that this was actually undertaken. What these were identified? Can we have some quotations related to such themes?

---

## Author Comment (AC2) · 24 Aug 2020

**This is a valuable and well-written submission. It tackles an important issue and makes good links with the existing literature; the analysis is excellent and the findings add considerably to what is already known in the published literature.**

We thank Prof Reiss for taking the time to review the manuscript and for their assessment of its quality.

**There is a degree of self-congratulation in the comparisons with other pro-**

[Figure]

**grammes – but the comparisons are very interesting!**

We have limited comparisons between PRiSE and other similar programmes merely to data about the schools involved as well as to the national statistics. Our aim was to objectively present any significant differences in these data and critically reflect on them, for example we note in the manuscript required improvements in PRiSE's targeting by school type and admissions policy in order to be more representative of all schools nationally, highlighting the policies enacted to help achieve this. However, if the reviewer has specific comments on phrasing that could be altered to mitigate a self-congratulatory tone then we would be happy to consider these.

**1. I have one major comment. It is a huge pity that "for ethical reasons we did not collect any protected characteristics (such as gender or race) or sensitive information (such as socio-economic background) from the students involved" (lines 67-68). Such data are not infrequently collected by educational researchers (indeed, they are collected by the DfE and available in the NPD) and I note the paragraph on gender that spans pages 7 and 8 (some might object to identifying gender in this way, though I am less of a purist). As the author is well aware, this means that all the conclusions made can only be made at school rather than individual student level. This, I am afraid, is not a trivial point. It is perfectly possible that the students who participate in these projects are far from representative of their schools. I think this should be made much clearer in the submission – in my view even the "School students from all backgrounds can do physics research" in the title is misleading and needs changed.**

We hope that the reviewer bears in mind that this evaluative work has resulted from a university department's schools engagement programme with limited resource that has been delivered and evaluated by physics researchers. It is therefore not an educational research project in and of itself and as such comes with many ethical and practical limitations. While educational researchers may be able to utilise data available in the UK Department for Education's National Pupil Database, it is somewhat impenetrable

in accessing even school census level data from a practitioners' perspective. Given these practicalities and the limited number of educational research studies into diversity and equity in STEM independent research projects at present, we felt that analysis even at the school-level would make a worthwhile contribution to the literature and in sharing good practice to other practitioners. However, we do take the reviewer's point that the manuscript could better flag the potential issue that PRiSE students may not be representative of their entire schools. We will therefore alter the title of the manuscript to "Schools of all backgrounds can do physics research", ensure phrasing throughout makes it clear our conclusions are limited to the school-level only, and expand the discussion justifying this school-level approach as follows:

This was done for both ethical and practical reasons, bearing in mind that this is a schools engagement programme delivered and evaluated by physics researchers and not an educational research project in and of itself. For example, it was deemed that requiring students or their teachers to provide protected or sensitive information upfront would have risked some students, or indeed entire schools, declining to participate. This limits the conclusions that can be made to only the school-level. However, it has been recognised that the clustering of students within schools results in students within the same school having more in common with each other than with students in different schools, an important consideration in the uptake of post-compulsory physics education for example (Gill and Bell, 2011). While multilevel models could account for this hierarchy, this is beyond the scope of what is practical for PRiSE. We note that schools typically involve entire (or significant fractions of) cohorts of A-level physics students in PRiSE (see M.O. Archer et al., 2020, for further discussion) and so while we have no indication that PRiSE students differ in any substantive way from their schools' wider student-base, we cannot rule out that they may not necessarily be representative. Finally, since one of the aims of PRiSE is to impact on

teachers' practice and schools' STEM environments, school-level considerations are valuable in this context regardless of the specific characteristics of the students engaged in PRiSE.

**2. I suspect the "issue" with IRIS is not in "their targeting" (line 155) but which schools respond to its offer**

We thank the reviewer for this perspective. While indeed the makeup of IRIS's schools may be simply due to those that respond to their offer, this somewhat passes the buck of the issue onto schools which is rather unfair. PRiSE and ORBYTS have demonstrated that there is interest in 'research in schools' projects from schools from a wide variety of backgrounds. Both these programmes have made considerations in developing their programmes to make them accessible for schools from a variety of backgrounds with the support provided, discussed further in M.O. Archer et al., 2020. Furthermore, PRiSE advertises via school networks (such as through the Institute of Physics and Ogden Trust) that specifically target disadvantaged schools, finding responses from a diversity of schools which is reflected in the statistics on participation. However, given that this brief comment positing potential reasons for the differences in schools engaging with IRIS and HiSPARC to those with PRiSE are merely conjecture, we would be amenable to remove them from the manuscript as they are not essential to its main messages and results that are rooted in objective data.

**3. I think it would be worth discussing briefly whether maximising retention of schools across years is always a good.**

The reviewer makes a good point which we will expand upon as follows:

> Maximising the retention of schools across years is necessarily a function of the capacity of a programme. While bringing new schools into the programme is certainly beneficial, we have seen that teachers' ability and confidence in supporting project work in their school develops not only across

the 6-month programme but over several years (M.O. Archer et al., 2020). Therefore, a practical balance could be to aim to involve schools directly for just a few years to the point that they can sustainably run projects with fewer interventions from the university, perhaps just an on campus kick-off and then the conference as researcher capacity is less of an issue with these. We acknowledge though that some schools might no longer participate without the full suite of interventions. This approach would enable the wider impacts on schools that benefit from multiple years of participation (M.O. Archer and DeWitt, 2020) while also ultimately freeing up capacity in the long-term for new schools to be able to benefit from the programme.

**4. Was there any ethical clearance for the research element of the work?**

Requirements for ethical clearance were discussed with an expert in ethics from Queen Mary's Joint Research Management Office. From these conversations it was deemed by them that the nature of this work (a schools engagement programme in physics delivered by physics researchers) and the ethical considerations put in place both within the programme and its evaluation (anonymisation of schools and participants, no protected/sensitive data being collected etc.), as well as the purpose of the publications being that of sharing practice with practitioners resulting from evaluative work were sufficient that Queen Mary did not require it to go before a formal ethics board.

**5. With reference to the qualitative data, there is a clear account of thematic analysis but then no evidence that this was actually undertaken. What these were identified? Can we have some quotations related to such themes?**

We apologise that the themes identified pertaining to accessibility, diversity and equity were not clear in the manuscript. We have now emboldened within the text the themes to clarify them, with the quotes within their respective paragraphs exemplifying the different aspects within these themes. The themes identified concerned:

- Valuing the diversity of the schools

- Equity in terms of students' ability

- Issues around communication

We hope that this is sufficient.

**References**

Archer, M. O., DeWitt, J., and Thorley, C.: Transforming school students' aspirations into destinations through extended interaction with cutting-edge research: Physics Research in School Environments, Geosci. Commun. Discuss., https://doi.org/10.5194/gc-2020-35, in review, 2020.

Archer, M. O. and DeWitt, J.: Thanks for helping me find my enthusiasm for physics! The lasting impacts research in schools projects can have on students, teachers, and schools, Geosci. Commun. Discuss., https://doi.org/10.5194/gc-2020-36, in review, 2020.

Gill, T. and Bell, J. F.: What Factors Determine the Uptake of A-level Physics?, International Journal of Science Education, 35, 753-772, doi: 10.1080/09500693.2011.577843, 2011.

---

## Referee Comment (RC2) · Anonymous Referee #2 · 1 Dec 2020

Overview

This paper and provides a more comprehensive and thorough analysis of a school engagement project than is typically done. The detail involved is welcomed, and it is excellent to see such evaluations being published.

Section 2 - Participation Regarding Independent schools who fail to initiate partnerships: were they allowed to continue with the programme? And were there any common reasons for not being able to set up partnerships? This information would be

useful for anyone trying to replicate the work.

Appendix A: how should a reader interpret the missing information about admissions policies? Is it simply not available publicly?

Background metrics: - It would be worth clarifying the difference between the full catchment area and the "local census area" for schools. My assumption is that it is just the area immediately surrounding the school's location/postcode - Thre is considerable variation between the local and full catchment data for indices of multiple deprivation, as noted in the text and the caption of Figure 3. The author discusses the difference in the data sources, though the text gives the impression that full catchment data is "better", when in fact the local data is in better agreement with the national average. - Is it due to the London/non-London locations, and if so does restricting the samples with local data to just those schools give more consistent results? In effect, are the differences because there are significant differences between the local/catchment data for a school, or is it because there is variation between the schools in areas where catchment data is/isn't available? (Or is there insufficient data to tell?!) - Another appropach may be to use national data for the relevant region (e.g. London/SE), but such data may not be available. - The clause "perhaps the issue lies in their targeting of and engagement with schools" regarding IRIS would benefit from clarification about how the approaches differ. It seems unbalanced to declare this as the issue with IRIS, but not HiSPAC. There are many, many other factors, such as the amount of teacher time required, the pupil time commitment etc., and it seems risky to attribute this to such a small subset. Perhaps wording along the line of "Such biases may be due to the cost of participation, the targeting of schools, or the engagement with schools, though it is noted that IRIS, like PRiSE, is free to schools".

Section 3 - Retention - It is mentioned in the text that participating in more than one year increases the retention of schools, and that the SCREAM data in Fig4a is perhaps an illustration of that. It would be interesting to see whether there is any more evidence of this - perhaps a similar plot to Fig 4a/b but with the data split by number of previous
years completed (perhaps 0 years / >=1 years).

Section 4 - Feedback - One of the reasons for teacher/school drop-off is the work required. What would be useful to see (either here or in the introduction) is an estimate of the role/tasks required of the teacher, or an estimate of the time involved. Experience (and the feedback in this paper) suggests that this is something that can have a big impact on school involvement.

---

## Author Comment (AC3) · 1 Dec 2020

**Overview - This paper and provides a more comprehensive and thorough analysis of a school engagement project than is typically done. The detail involved is welcomed, and it is excellent to see such evaluations being published.**

We thank the reviewer for their time in reviewing the manuscript and for their comments. We have considered each carefully.

**Section 2 - Participation Regarding Independent schools who fail to initiate part-**

[Figure]

**nerships: were they allowed to continue with the programme? And were there any common reasons for not being able to set up partnerships? This information would be useful for anyone trying to replicate the work.**

We thank the reviewer for this suggested inclusion. In the first instance of this policy (academic year 2019-2020) any school which refused to even try and form partnerships was not allowed on the programme, even if they had worked with us before. However, those schools which had agreed to the policy but then failed to form a partnership were allowed to participate, with the expectation that they try again to form such a partnership for participation a year later. Typical reasons for this failure were not being able to draw from existing local partnerships, limited time from the application to the summer holidays, and poor communication between teachers at different schools. We will add these points to the manuscript.

**Appendix A: how should a reader interpret the missing information about admissions policies? Is it simply not available publicly?**

The reviewer is correct, missing information is due to it not being publicly available. We will make a note of this in the revision.

**Background metrics: - It would be worth clarifying the difference between the full catchment area and the "local census area" for schools. My assumption is that it is just the area immediately surrounding the school's location/postcode**

The reviewer is correct, we will add this clarification using the reviewer's suggested wording.

**There is considerable variation between the local and full catchment data for indices of multiple deprivation, as noted in the text and the caption of Figure 3. The author discusses the difference in the data sources, though the text gives the impression that full catchment data is "better", when in fact the local data is in better agreement with the national average.**

We believe that full catchment area data is more reflective of a school's student base as schools will draw students from a wider range of locations than simply the census area within which they are located, thus taking account of the full range of locations will yield more reflective metrics about a school's students. We will add this point to the manuscript.

**- Is it due to the London/non-London locations, and if so does restricting the samples with local data to just those schools give more consistent results? In effect, are the differences because there are significant differences between the local/catchment data for a school, or is it because there is variation between the schools in areas where catchment data is/isn't available? (Or is there insufficient data to tell?!)**

There are significant differences between local and catchment data in general. This was investigated in Appendix C, where we compared the local and catchment data for all schools in London (the only schools for which both sets of data are publicly available). This showed that while the local and catchment data certainly correlate, this correlation is not particularly strong (coefficients ranging from 0.64-0.85). We will make a reference to this fact in the main text.

**Another approach may be to use national data for the relevant region (e.g. London/SE), but such data may not be available.**

We did consider this. However, we felt that benchmarking against only London data would have implications on how we compare PRiSE to the other programmes, which are not located in London. Including national data as well as data across all of London (both local and catchment) on Figure 3 would make it far too busy. For simplicity, we therefore decided to only benchmark against national data.

**The clause "perhaps the issue lies in their targeting of and engagement with schools" regarding IRIS would benefit from clarification about how the approaches differ. It seems unbalanced to declare this as the issue with IRIS, but**

**not HiSPAC. There are many, many other factors, such as the amount of teacher time required, the pupil time commitment etc., and it seems risky to attribute this to such a small subset. Perhaps wording along the line of "Such biases may be due to the cost of participation, the targeting of schools, or the engagement with schools, though it is noted that IRIS, like PRiSE, is free to schools".**

We have decided to remove these conjectures about other programmes from the manuscript as they are not vital to the results presented.

**Section 3 - Retention - It is mentioned in the text that participating in more than one year increases the retention of schools, and that the SCREAM data in Fig4a is perhaps an illustration of that. It would be interesting to see whether there is any more evidence of this - perhaps a similar plot to Fig 4a/b but with the data split by number of previous years completed (perhaps 0 years / >=1 years).**

The reviewer raises an interesting point. We will add further evidence of this from data given in Appendix A. In the attached figure, the top panel shows a scatter plot (blue) of the number of years each school was involved and the number of years they completed. Schools tend to lie near the two possible extremes (solid lines) of either completing the projects every year they were involved or not completing the projects at all. This is exemplified in the bottom panel, showing the distribution of the proportion of years that schools completed. This has a clear bimodal form peaked close to zero and one. Another point is that in the top panel, as the number of years involved increase the number of years completed tends towards the positive extreme further. This is shown most clearly in the red plot which shows the mean number of years completed and its standard error as a function of number of years involved. We will add this figure and these points to the paper.

**Section 4 - Feedback - One of the reasons for teacher/school drop-off is the work required. What would be useful to see (either here or in the introduction) is an estimate of the role/tasks required of the teacher, or an estimate of the time**

Interactive
comment

**involved. Experience (and the feedback in this paper) suggests that this is something that can have a big impact on school involvement.**

A discussion of the role of the teacher in assisting with these projects is given in more detail in the paper introducing the framework (M.O. Archer et al., 2020). However, we can add a note to the introduction highlighting the key points of this discussion. Under PRiSE the teacher's role is chiefly one of encouraging their students to persist, providing what advice they can, and then communicating with the university. Teachers are not expected to fully manage the projects, which is why numerous modes of support are provided from active researchers who have the expertise and skills in the areas of each project.

Archer, M. O., DeWitt, J., and Thorley, C.: Transforming school students' aspirations into destinations through extended interaction with cutting-edge research: Physics Research in School Environments, Geosci. Commun. Discuss., https://doi.org/10.5194/gc-2020-35, in review, 2020.

[Figure]

**Fig. 1.** Years completed against years involved

---

## Author Response (AR1)

**Response to Reviewers**

School students from all backgrounds can do physics research: On the accessibility and equity of the PRiSE approach to independent research projects
Archer

We thank the editor and the reviewers for their comments. We have revised the manuscript in response to these, which we detail here. Line numbers refer to the tracked changes version of the manuscript.

**Editor comments**

**Apologies for the delay in following up on the referee reports and your subsequent comments, but I am keen to wrap this paper up given that both reviews are positive and proposed revisions are minor.**

**Having reading the ms and your considered responses to the both reviewers, I'd appreciate it if you could specifically address the following points in making minor revisions to your manuscript:**

**(1) more prominently highlight the potential issue that PRiSE students may not be representative of their entire schools and accordingly alter the title of the manuscript as you have proposed;**

We have adjusted the title, changed any mention that results relate to all school students, and have added further discussion of our school-level approach and that PRiSE students may not necessarily be representative on lines 76-88.

**(2) add a very brief comment to clarify the nature of the thematic analysis.**

We clarify the thematic analysis on lines 315-316.

**(3) acknowledge the criticism that citizen-science projects may often not be sufficiently audience-centric by more explicitly directing the reader to Archer et al., 2020.**

This point has been added to lines 32-35.

**I am happy for you to amend the revised manuscript to acknowledge and clarify other points raised by Prof Reiss, namely relating to the conjecture around the schools engaging with IRIS and HiSPARC and to the issue of retention.**

Conjectures about other programmes have been removed (see lines 184-186) and we raise the need for further specific qualitative research to understand participation on lines 93-95. Retention across years is further discussed on lines 286-293.

**With regard other points raised, I personally did not recognise the tone of the paper as self-contratulatory and the ethical clarification is now on record.**

**I trust that these changes will be relatively easy to undertake - it seems like you have already revised the ms with them in mind. So on that basis I look forward to receiving a copy of the final submission.**

We thank the editor for these comments.

**RC1**

**This is a valuable and well-written submission. It tackles an important issue and makes good links with the existing literature; the analysis is excellent and the findings add considerably to what is already known in the published literature.**

We thank Prof Reiss for taking the time to review the manuscript and for their assessment of its quality.

**There is a degree of self-congratulation in the comparisons with other programmes – but the comparisons are very interesting!**

We have limited comparisons between PRiSE and other similar programmes merely to data about the schools involved as well as to the national statistics.  Our aim was to objectively present any significant differences in these data and critically reflect on them, for example we note in the manuscript required improvements in PRiSE's targeting by school type and admissions policy in order to be more representative of all schools nationally, highlighting the policies enacted to help achieve this.

**1. I have one major comment. It is a huge pity that "for ethical reasons we did not collect any protected characteristics (such as gender or race) or sensitive information (such as socio-economic background) from the students involved" (lines 67-68). Such data are not infrequently collected by educational researchers (indeed, they are collected by the DfE and available in the NPD) and I note the paragraph on gender that spans pages 7 and 8 (some might object to identifying gender in this way, though I am less of a purist). As the author is well aware, this means that all the conclusions made can only be made at school rather than individual student level. This, I am afraid, is not a trivial point. It is perfectly possible that the students who participate in these projects are far from representative of their schools. I think this should be made much clearer in the submission – in my view even the "School students from all backgrounds can do physics research" in the title is misleading and needs changed.**

We hope that the reviewer bears in mind that this evaluative work has resulted from a university department's schools engagement programme with limited resource that has been delivered and evaluated by physics researchers. It is therefore not an educational research project in and of itself and as such comes with many ethical and practical limitations. While educational researchers may be able to utilise data available in the UK Department for Education's National Pupil Database, it is somewhat impenetrable in accessing even school census level data from a practitioners' perspective. Given these practicalities and the limited number of educational research studies into diversity and equity in STEM independent research projects at present, we felt that analysis even at the school-level would make a worthwhile contribution to the literature and in sharing good practice to other practitioners. However, we do take the reviewer's point that the manuscript could better flag the potential issue that PRiSE students may not be representative of their entire schools. We have therefore altered the title of the manuscript to "Schools of all backgrounds can do physics research", ensured phrasing throughout makes it clear our conclusions are limited to the school-level only, and expand the discussion justifying this school-level approach on lines 76-88.

**2. I suspect the "issue" with IRIS is not in "their targeting" (line 155) but which schools respond to its offer**

We thank the reviewer for this perspective. While indeed the makeup of IRIS's schools may be simply due to those that respond to their offer, this somewhat passes the buck of the issue

onto schools which is rather unfair. PRiSE and ORBYTS have demonstrated that there is interest in 'research in schools' projects from schools from a wide variety of backgrounds. Both these programmes have made considerations in developing their programmes to make them accessible for schools from a variety of backgrounds with the support provided, discussed further in M. O. Archer et al., 2020. Furthermore, PRiSE advertises via school networks (such as through the Institute of Physics and Ogden Trust) that specifically target disadvantaged schools, finding responses from a diversity of schools which is reflected in the statistics on participation. However, given that this brief comment positing potential reasons for the differences in schools engaging with IRIS and HiSPARC to those with PRiSE are merely conjecture, we have removed them from the manuscript as they are not essential to its main messages and results that are rooted in objective data.

**3. I think it would be worth discussing briefly whether maximising retention of schools across years is always a good.**

The reviewer makes a good point which have expanded the discussion on lines 286-293.

**4. Was there any ethical clearance for the research element of the work?**

Requirements for ethical clearance were discussed with an expert in ethics from Queen Mary's Joint Research Management Office. From these conversations it was deemed by them that the nature of this work (a schools engagement programme in physics delivered by physics researchers) and the ethical considerations put in place both within the programme and its evaluation (anonymisation of schools and participants, no protected/sensitive data being collected etc.), as well as the purpose of the publications being that of sharing practice with practitioners resulting from evaluative work were sufficient that Queen Mary did not require it to go before a formal ethics board.

**5. With reference to the qualitative data, there is a clear account of thematic analysis but then no evidence that this was actually undertaken. What these were identified? Can we have some quotations related to such themes?**

We apologise that the themes identified pertaining to accessibility, diversity and equity were not clear in the manuscript. We have now emboldened within the text the themes to clarify them, with the quotes within their respective paragraphs exemplifying the different aspects within these themes. The themes identified concerned:

- Valuing the diversity of the schools
- Equity in terms of students' ability
- Issues around communication

This has been noted on lines 315-316.

**RC2**

**Overview**

**This paper and provides a more comprehensive and thorough analysis of a school engagement project than is typically done. The detail involved is welcomed, and it is excellent to see such evaluations being published.**

We thank the reviewer for their time in reviewing the manuscript and for their comments. We have considered each carefully.

**Section 2 - Participation Regarding Independent schools who fail to initiate partnerships: were they allowed to continue with the programme? And were there any common reasons for not being able to set up partnerships? This information would be useful for anyone trying to replicate the work.**

We thank the reviewer for this suggested inclusion. In the first instance of this policy (academic year 2019-2020) any school which refused to even try and form partnerships was not allowed on the programme, even if they had worked with us before. However, those schools which had agreed to the policy but then failed to form a partnership were allowed to participate, with the expectation that they try again to form such a partnership for participation a year later. Typical reasons for this failure were not being able to draw from existing local partnerships, limited time from the application to the summer holidays, and poor communication between teachers at different schools. These points have been added on lines 115-120.

**Appendix A: how should a reader interpret the missing information about admissions policies? Is it simply not available publicly?**

The reviewer is correct, missing information is due to it not being publicly available. We now make a note of this.

**Background metrics: - It would be worth clarifying the difference between the full catchment area and the "local census area" for schools. My assumption is that it is just the area immediately surrounding the school's location/postcode**

The reviewer is correct, we have used the reviewer's suggested wording (lines 124-125).

**There is considerable variation between the local and full catchment data for indices of multiple deprivation, as noted in the text and the caption of Figure 3. The author discusses the difference in the data sources, though the text gives the impression that full catchment data is "better", when in fact the local data is in better agreement with the national average.**

We believe that full catchment area data is more reflective of a schools' student base as schools will draw students from a wider range of locations than simply the census area within which they are located, thus taking account of the full range of locations will yield more reflective metrics about a school's students. This has been added on lines 125-126.

**- Is it due to the London/non-London locations, and if so does restricting the samples with local data to just those schools give more consistent results? In effect, are the differences because there are significant differences between the local/catchment data for a school, or is it because there is variation between the schools in areas where catchment data is/isn't available? (Or is there insufficient data to tell?!)**

There are significant differences between local and catchment data in general. This was investigated in Appendix C, where we compared the local and catchment data for all schools in London (the only schools for which both sets of data are publicly available). This showed that while the local and catchment data certainly correlate, this correlation is not particularly strong (coefficients ranging from 0.64-0.85). We have expanded this analysis in the appendix, demonstrating that the while the distributions across all London schools remain similar across both datasets, individual schools can result in significantly different values depending on which is used. This is also pointed out in the main text on lines 126-128.

**Another approach may be to use national data for the relevant region (e.g. London/SE), but such data may not be available.**

We did consider this. However, we felt that benchmarking against only London data would have implications on how we compare PRiSE to the other programmes, which are not located in London. Including national data as well as data across all of London (both local and catchment) on Figure 3 would make it far too busy. For simplicity, we therefore decided to only benchmark against national data.

**The clause "perhaps the issue lies in their targeting of and engagement with schools" regarding IRIS would benefit from clarification about how the approaches differ. It seems unbalanced to declare this as the issue with IRIS, but not HiSPAC. There are many, many other factors, such as the amount of teacher time required, the pupil time commitment etc., and it seems risky to attribute this to such a small subset. Perhaps wording along the line of "Such biases may be due to the cost of participation, the targeting of schools, or the engagement with schools, though it is noted that IRIS, like PRiSE, is free to schools".**

We have decided to remove these conjectures about other programmes from the manuscript as they are not vital to the results presented.

**Section 3 - Retention - It is mentioned in the text that participating in more than one year increases the retention of schools, and that the SCREAM data in Fig4a is perhaps an illustration of that. It would be interesting to see whether there is any more evidence of this - perhaps a similar plot to Fig 4a/b but with the data split by number of previous years completed (perhaps 0 years / >=1 years).**

The reviewer raises an interesting point. We have added further evidence of this from data given in Appendix A. Figure 6 shows a scatter plot (blue) of the number of years each school was involved and the number of years they completed. Discussion of this figure can be found on lines 262-272, further backing up the original point.

**Section 4 - Feedback - One of the reasons for teacher/school drop-off is the work required. What would be useful to see (either here or in the introduction) is an estimate of the role/tasks required of the teacher, or an estimate of the time involved. Experience (and the feedback in this paper) suggests that this is something that can have a big impact on school involvement.**

A discussion of the role of the teacher in assisting with these projects is given in more detail in the paper introducing the framework (M.O. Archer et al., 2020). However, we add a note to the introduction highlighting the key points of this discussion. This can be found on lines 41-44.

**SC1**

**One thing I have noticed is principle investigators on research projects using citizen scientists as cheap labor. By that I mean they advertise for people to contribute and then instruct them to do menial tasks such as data transcription and collation, especially in climate science. IMO, this is the last thing that you want students to be doing – expect much greater things from them. Let them work the algorithms and mathematical physics and encourage them to find the next great ansatz that might lead to a research breakthrough. That's all I have to say, because if history is any indication, insight can come from anywhere.**

We agree with this comment that some citizen science projects are not sufficiently audience-focused to give them a meaningful experience of interacting with the research. The PRiSE approach,

however, is very different to this, with the participants gaining an authentic research experience being of primary importance. This is discussed in light of current citizen science practices in further detail both in the companion to this paper (Archer et al., 2020) as well as an earlier paper for one of the PRiSE projects MUSICS (Archer et al., 2018). We have added a note on this here on lines 32-35.

**References**

Archer, M. O., Hartinger, M. D., Redmon, R., Angelopoulos, V., Walsh, B. M., & Eltham Hill School Year 12 Physics students: First results from sonification and exploratory citizen science of magnetospheric ULF waves: Long-lasting decreasing-frequency poloidal field line resonances following geomagnetic storms. Space Weather, 16, 1753– 1769. https://doi.org/10.1029/2018SW001988, 2018.

Archer, M. O., DeWitt, J., and Thorley, C.: Transforming school students' aspirations into destinations through extended interaction with cutting-edge research: Physics Research in School Environments, Geosci. Commun. Discuss., https://doi.org/10.5194/gc-2020-35, in review, 2020.